# CHG Shapley: Efficient Data Valuation and Selection towards Trustworthy Machine Learning

## Abstract

Understanding the decision-making process of machine learning models is crucial for ensuring trustworthy machine learning. Data Shapley, a landmark study on data valuation, advances this understanding by assessing the contribution of each datum to model performance. However, the resource-intensive and time-consuming nature of multiple model retraining poses challenges for applying Data Shapley to large datasets. To address this, we propose the CHG (compound of Hardness and Gradient) utility function, which approximates the utility of each data subset on model performance in every training epoch. By deriving the closed-form Shapley value for each data point using the CHG utility function, we reduce the computational complexity to that of a single model retraining, achieving a quadratic improvement over existing marginal contribution-based methods. We further leverage CHG Shapley for real-time data selection, conducting experiments across three settings: standard datasets, label noise datasets, and class imbalance datasets. These experiments demonstrate its effectiveness in identifying high-value and noisy data. By enabling efficient data valuation, CHG Shapley promotes trustworthy model training through a novel data-centric perspective.

## 1 Introduction

The central problem of trustworthy machine learning is explaining the decision-making process of models to enhance the transparency of data-driven algorithms. However, the high complexity of machine learning model training and inference processes obscures an intuitive understanding of their internal mechanisms. Approaching trustworthy machine learning from a data-centric perspective (Liu et al., 2023) offers a new perspective for research. For trustworthy model inference, a representative algorithm is the SHAP (Lundberg & Lee, 2017), which quantitatively attributes model outputs to input features, clarifying which features influence specific results the most. SHAP and its variants(Kwon & Zou, 2022b) are widely applied in data analysis and healthcare. For trustworthy model training, the Data Shapley algorithm (Ghorbani & Zou, 2019) stands out. It quantitatively attributes a model's performance to each training data point, identifying valuable data that improves performance and noisy data that degrades it. Data valuation reveals how much each training sample affects model performance, serving as a foundation for tasks like data selection, acquisition, and cleaning, while also facilitating the creation of data markets (Mazumder et al., 2023).

The impressive effectiveness of both SHAP and Data Shapley algorithms is rooted in the Shapley value (Shapley, 1953). The unique feature of the Shapley value lies in its ability to accurately and fairly allocate contributions to each factor in decision-making processes where multiple factors interact with each other. However, the exact computation of the Shapley value is $\mathcal{O}(2^n)$, where $n$ is the number of factors. Even with estimation techniques such as linear least squares regression (Lundberg & Lee, 2017) or Monte Carlo methods (Ghorbani & Zou, 2019), the efficiency of SHAP and Data Shapley algorithms struggles to scale with high-dimensional inputs or large datasets.

In this paper, we focus on the efficiency problem of data valuation on large-scale datasets. Instead of investigating more efficient and robust algorithms to approximate the Shapley value as in (Lundberg & Lee, 2017; Wang et al., 2024b), we concentrate on estimating the utility function. The key intuition is that if the performance of a model trained on a data subset can be expressed analytically,

a closed-form solution for each datum's Shapley value could be derived, offering a more computationally efficient solution. Our contributions are summarized as follows:

**Efficient and Large-Scale Data Valuation Method**: We introduce the CHG (Compound of Hardness and Gradient) score, which assesses the influence of a data subset on model accuracy. By deriving the analytical expression for the Shapley value of each data point under this utility function, we significantly improve upon the $\mathcal{O}(n^2 \log n)$ time complexity of Data Shapley to a single model training run.

**Real-Time Training Data Selection in Large Datasets**: Calculating data value was considered computationally intensive, making real-time data selection based on data value impractical. Due to the efficient computation of CHG Shapley, we further employ it for real-time data selection. Experiments conducted across three settings—standard datasets, label noise datasets, and class imbalance datasets—demonstrate CHG Shapley's effectiveness in identifying high-value and noisy data.

**A New Data-Centric Perspective on Trustworthy Machine Learning**: In real-world scenarios, data may not only be subject to noise and class imbalance, but it can also involve more complex mixtures of these issues. CHG Shapley is a parameter-free method that derives its advantages solely from a deeper understanding of the data. We believe that the proposed method, grounded in data valuation, has the potential to enhance trustworthy model training in complex data distributions.

## 2 BACKGROUND AND RELATED WORKS

**Data-centric AI**. For an extended period, the machine learning research community has predominantly concentrated on model development rather than on the underlying datasets (Mazumder et al., 2023). However, the inaccuracies, unfairness, and biases in models caused by data have become increasingly severe in real-world applications (Wang et al., 2023). Conducting trustworthy machine learning research from a data-centric perspective has garnered increasing attention from researchers (Liu et al., 2023). Research on data valuation can also be seen as an effort to understand the machine learning model training process from a data-centric perspective.

**Shapley value**. The Shapley value (Shapley, 1953) is widely regarded as the fairest method for allocating contributions (Rozemberczki et al., 2022; Algaba et al., 2019). Given a set of players $N$ and a utility function $U$ defined on subsets of $N$, the Shapley value is the only solution that satisfies the axioms of dummy player, symmetry, efficiency, and linearity:

- Dummy player: If $U(S \cup \{i\}) = U(S)$ for all $S \subseteq N \setminus \{i\}$, then $\phi(i; U) = 0$.
- Symmetry: If $U(S \cup \{i\}) = U(S \cup \{j\})$ for all $S \subseteq N \setminus \{i, j\}$, then $\phi(i; U) = \phi(j; U)$.
- Linearity: For utility functions $U_1, U_2$ and any $\alpha_1, \alpha_2 \in \mathbb{R}$, $\phi(i; \alpha_1 U_1 + \alpha_2 U_2) = \alpha_1 \phi(i; U_1) + \alpha_2 \phi(i; U_2)$.
- Efficiency: $\sum_{i \in N} \phi(i; U) = U(N) - U(\emptyset)$.

**Definition 1 (Shapley (1953))** *Given a player set $N$ and a utility function $U$, the Shapley value of a player $i \in N$ is defined as*

$$\phi_i(U) := \frac{1}{n} \sum_{k=1}^{n} \binom{n-1}{k-1}^{-1} \sum_{S \subseteq N \setminus \{i\}, |S| = k-1} [U(S \cup i) - U(S)] \tag{1}$$

A more intuitive form for the Shapley value is:

$$\phi_i(U) = \mathbb{E}_{\pi \sim \Pi} \left[ U\left( S_\pi^i \cup \{i\} \right) - U\left( S_\pi^i \right) \right]. \tag{2}$$

Here, $\pi \sim \Pi$ refers to a uniformly random permutation of the player set $N$, and $S_\pi^i$ denotes the set of players preceding player $i$ in the permutation $\pi$. This expression highlights that the Shapley value for player $i$ captures the expected marginal contribution of that player across all possible subsets. Fig. 1 illustrates the process of computing Shapley values.

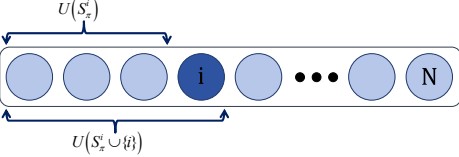

Figure 1: Shapley value's calculation.

**Data valuation**. Data valuation aims to quantitatively analyze the impact of training data on the performance of machine learning models, particularly deep neural networks. Determining the value of data can facilitate tasks such as data selection (Wang et al., 2024c), acquisition(Mazumder et al., 2023), and trading(Wang et al., 2024a). Since the performance of machine learning models is derived from the collective contribution of all data points, a fair data valuation algorithm is essential to accurately attribute each data point's contribution to the model's performance.

A milestone in data valuation research is the Data Shapley algorithm developed by Ghorbani & Zou (2019). Data Shapley builds upon the Shapley value formula by treating all data in the training dataset as players in a cooperative game of "training the model together". $U(S)$ represents the accuracy achieved by training the model on dataset subset $S$. Consequently, the Shapley value computed for each data point serves as an indication of its contribution to model performance. Data points with lower values are considered more likely to be harmful, while those with higher values are deemed to be lacking in the dataset. Removing low-value or acquiring more high-value data can improve accuracy. Following Data Shapley, several subsequent have been proposed.

KNN Shapley (Jia et al., 2019) focuses on the Shapley value for k-nearest neighbors, offering a closed-form solution that improves computational efficiency to $\mathcal{O}(\sqrt{n}\log(n)^2)$. Beta Shapley (Kwon & Zou, 2022a) addresses the uniform weighting issue in Data Shapley by relaxing the efficiency axiom and using a beta distribution, improving performance in tasks like mislabeled data detection. Data Banzhaf (Wang & Jia, 2023) reduces noise in model training by utilizing the Banzhaf value, shown to be robust in noisy environments. LAVA (Just et al., 2023) developed a proxy for validation performance, enabling the evaluation of data in a manner that is agnostic to the learning algorithms. DVRL (Kwon & Zou, 2023) and Data-OOB (Yoon et al., 2020) utilize reinforcement learning and out-of-bag estimation, respectively, to assess data values.

Several other works use gradient information to compute the value of training data. Xu et al. (2021) propose the cosine GradE Shapley value (CGSV) to measure each agent's contribution in a collaborative machine learning scenario. CGSV approximates an agent's Shapley value during the aggregation process by using the cosine similarity between the agent's gradient and the aggregated gradient of all agents. Pruthi et al. (2020) introduced TracIn, a method that measures the influence of a training example on a test example by computing the inner product between their respective gradients.

Finally, Jiang et al. (2023) introduced the OpenDataVal benchmark, which focuses on data valuation algorithm design by abstracting model training. We also use this benchmark to evaluate our proposed algorithm in data valuation experiments.

**Data selection**. There are two main streams in the data selection field. The first focuses on determining the scores of individual data points, employing methods such as influence functions (Koh & Liang, 2017; Yang et al., 2023; Chhabra et al., 2024), the EL2N score (Paul et al., 2021), and loss (i.e., hardness) metrics (e.g., InfoBatch (Qin et al., 2024), worst-case training (Huang et al., 2022)). These approaches typically use the information from a single data point and model parameters to calculate scores for that point, subsequently selecting the data points with the highest scores. Recently, Xia et al. (2024) proposed LESS, which sets the gradient mean of data within a specific downstream task as the target gradient and employs the inner product or cosine similarity to select data with the most similar gradients from a large collection of instruction datasets for model fine-tuning. Their experiments (Xia et al., 2024) show that training on just 5% of the data selected by LESS often outperforms training on the full dataset across various downstream tasks.

The second approach, including coreset methods such as GradMatch (Killamsetty et al., 2021a) and Glister (Killamsetty et al., 2021b), as well as data distillation (Zhao et al., 2021), emphasizes the utility of data subsets. These methods aim to identify a subset that effectively represents the original dataset while achieving comparable model accuracy. They typically leverage the gradients of individual training data points to form a subset whose gradient direction aligns with that of the validation set.

Our data selection method, CHG Shapley, which is grounded in data valuation principles, represents a fusion of these two streams: it leverages the utility of different training subsets to calculate a score (i.e., Shapley value) for each individual data point. We will compare our method with these two approaches in data selection experiments, demonstrating its ability to identify high-value (i.e., representative) data and low-value (i.e., noisy) data.

# 3 Efficient Data valuation

## 3.1 Preliminaries

Let $N = \{1, \ldots, n\}$ denote a labeled training set. The objective is to assign a scalar value to each training data point based on its contribution to the model's performance. A utility function $U : 2^N \to \mathbb{R}$ maps any subset $S$ of the training set (including the empty set) to the performance of the model trained on that subset, denoted as $U(S)$. In Data Shapley (Ghorbani & Zou, 2019), the model's accuracy on a hold-out test set is utilized as $U(S)$. Besides, similar to Data Shapley, we do not impose any distributional assumptions on the data, and independence among the data points is not required.

## 3.2 CHG Score as a Utility Function for Subset Model Performance

The opacity of $U(S)$, or the fact that $U(S)$ can only be obtained after training a model on $S$, renders marginal contribution-based data valuation algorithms such as Data Shapley (Ghorbani & Zou, 2019), Beta Shapley (Kwon & Zou, 2022a), Data Banzhaf (Wang & Jia, 2023), AME (Lin et al., 2022), and LOO (leave-one-out error) time-consuming and computationally intensive. On the contrary, if we can determine the model's performance on $S$ without training on it (Wu et al., 2022; Ki et al., 2023), or even derive the expression for $U(S)$ in terms of $S$, then we may significantly reduce the model training time and expedite the calculation of each data's value. To achieve this, we first investigate the impact of training the model using a subset of data $S$.

The objective of model training is to minimize training loss while ensuring generalization (Zhang et al., 2017) through an appropriate dataset, sufficient data quantity, model complexity, and regularization techniques, etc. Once the dataset, model architecture, regularization techniques, and other methods that ensure generalization are established, the goal with all training data is to minimize training loss as much as possible. Therefore, in our study, the utility function is defined as the degree of decrease in training loss.

Let the loss function of the neural network across the entire training dataset be represented as $f(\theta) := \frac{1}{N} \sum_{i \in N} f(i; \theta)$. When training the model on a subset $S$ using gradient descent, the next update direction for the parameters is given by $x := \frac{1}{|S|} \sum_{i \in S} x_i$, where $x_i$ represents the gradient of the $i$-th data point. After one step of gradient descent, the parameters are updated as $\theta \leftarrow \theta - \eta x$. Based on the following lemma, the term $\|\nabla f(\theta)\|_2^2 - \|\nabla f(\theta) - x\|_2^2$ can be employed as a utility function to evaluate the impacts of the subset $S$ on the training loss.

**Lemma 1 ((Nesterov, 2018))** *Let $f$ be any differentiable function with L-Lipschitz continuous gradient, $\theta_x = \theta - \eta x$, and $\eta = 1/L$[1], then we have:*

$$f(\theta_x) \leq f(\theta) - \frac{1}{2L}(\|\nabla f(\theta)\|_2^2 - \|\nabla f(\theta) - x\|_2^2) \tag{3}$$

*Proof: Due to the definition of L-Lipschitz continuity, We have $f(\theta_x) \leq f(\theta) + \langle \nabla f(\theta), \theta_x - \theta \rangle + \frac{L}{2}\|\theta_x - \theta\|_2^2$. Plugging in $\theta_x = \theta - \eta x$ to it, we get $\langle \nabla f(\theta), \theta_x - \theta \rangle = \langle \nabla f(\theta), -\eta x \rangle$ and $\|\theta_x - \theta\|_2^2 = \eta^2 \|x\|_2^2$. And with $\eta = 1/L$, we then have $f(\theta) + \langle \nabla f(\theta), \theta_x - \theta \rangle + \frac{L}{2}\|\theta_x - \theta\|_2^2 = f(\theta) + \langle \nabla f(\theta), -\eta x \rangle + \frac{L}{2}\eta^2\|x\|_2^2 = f(\theta) - \frac{1}{2L}(2\langle \nabla f(\theta), x \rangle - \|x\|_2^2) = f(\theta) - \frac{1}{2L}(\|\nabla f(\theta)\|_2^2 - \|\nabla f(\theta) - x\|_2^2)$.*

Lemma 1 is particularly important to our whole method. It suggests that the Euclidean distance of two gradients is not just a similarity metric like inner product or cosine, but also gives an upper bound of the training loss, which then serves as the utility function of any data subset to calculate the Shapley value of each single data.

In general, when there is no label noise, focusing on hard-to-learn data points can enhance the model's generalization ability (Huang et al., 2022) and reduce training time without affecting test performance (Qin et al., 2024). Thus, we propose to incorporate the hardness of data points into the

---

[1]This is a commonly used assignment method, for example, in proving the convergence of gradient descent for non-convex functions to local minima (Nesterov, 2018), $\eta$ was set as $1/L$.

| Algorithms | Underlying Method | Model Retraining Complexity | Quality of Data Values | | | |
|---|---|---|---|---|---|---|
| | | | Noisy Label Detection | Noisy Feature Detection | Point Removal | Point Addition |
| LOO | Marginal contribution | $n$ | - | - | + | + |
| Data Shapley (Ghorbani & Zou, 2019) | Marginal contribution | $\mathcal{O}(n^2 \log n)$ | + | + | ++ | ++ |
| KNN Shapley (Jia et al., 2019) | Marginal contribution | NA | + | + | ++ | ++ |
| Beta Shapley (Kwon & Zou, 2022a) | Marginal contribution | $\mathcal{O}(n^2 \log n)$ | + | + | ++ | ++ |
| Data Banzhaf (Wang & Jia, 2023) | Marginal contribution | $\mathcal{O}(n \log n)$ | - | - | + | + |
| AME (Lin et al., 2022) | Marginal contribution | $\mathcal{O}(n^2 \log n)$ | - | - | ++ | + |
| Influence Function (Koh & Liang, 2017) | Gradient | NA | - | - | + | + |
| LAVA (Just et al., 2023) | Gradient | NA | - | ++ | + | + |
| DVRL (Yoon et al., 2020) | Importance weight | 1 | + | - | + | ++ |
| Data-OOB (Kwon & Zou, 2023) | Out-of-bag estimate | NA | ++ | + | - | ++ |
| CHG Shapley (this paper) | Marginal contribution, Gradient and Loss | 1 | ++ | ++ | ++ | NA |

Table 1: A taxonomy of data valuation algorithms, mainly abstracted from (Jiang et al., 2023). The symbol "NA" means the method is not based on model training, then "No Answer". The symbols '- / + / ++' indicate that a corresponding data valuation method achieves a 'similar or worse / better / much better' performance than a random baseline, respectively. Detailed experimental results comparing these data valuation methods are provided in Appendix A.

model optimization process when there is little to no label noise. Specifically, our optimization objective is to minimize $\frac{1}{N} \sum_{i \in N} h_i f(i; \theta)$ rather than usual empirical risk $\frac{1}{N} \sum_{i \in N} f(i; \theta)$, where $h_i$ is equal to $f(i; \theta)$ but does not participate in model backpropagation. Here, $h_i$ can also be interpreted as an unnormalized probability of selecting data point $i$ based on its hardness. Then the gradient of $i$-th data changes from $x_i$ to $h_i x_i$, and then we define the Compound of Hardness and Gradient score (CHG score) of a data subset $S$ as $U(S) := \| \frac{1}{N} \sum_{i \in N} h_i \nabla f(i; \theta) \|_2^2 - \| \frac{1}{N} \sum_{i \in N} h_i \nabla f(i; \theta) - \frac{1}{S} \sum_{i \in S} h_i \nabla f(i; \theta) \|_2^2$. When the subset $S = \emptyset$, the produced utility $U(S)$ is 0.

However, when label noise cannot be ignored, the hardness metric becomes unreliable, as a well-performing classifier may assign a high hardness value to a training data point with an incorrect label. In this case, we still utilize the original score $\| \frac{1}{N} \sum_{i \in N} \nabla f(i; \theta) \|_2^2 - \| \frac{1}{N} \sum_{i \in N} \nabla f(i; \theta) - \frac{1}{S} \sum_{i \in S} \nabla f(i; \theta) \|_2^2$, the corresponding method for calculating the Shapley value based on Equation 4 is referred to as **GradE Shapley** (Gradient Euclidean Shapley). We also denote the direct use of the hardness $h_i$ as the Shapley value of data point $i$ as **Hardness Shapley**. We compared these three methods across various dataset settings in our data selection experiments.

## 3.3 CHG Shapley-based Data Valuation Algorithm

After obtaining the CHG score, which approximates the influence of data subsets on training loss using gradient and loss information during model training, we can derive the analytical expression for the Shapley value of each data point under this utility function. This method, denoted as **CHG Shapley**, enables efficient computation of the value of each data point in large-scale datasets.

**Theorem 1** *Supposing for any subset $S \subseteq N$, $U(S) = \|\alpha\|_2^2 - \| \frac{1}{S} \sum_{i \in S} x_i - \alpha \|_2^2$, then the Shapley value of $j$-th data point can be expressed as:*

$$\phi_j(U) =$$
$$\left( -\frac{\sum_{k=1}^n \frac{1}{k^2}}{n} + \frac{2 \sum_{k=1}^n \frac{1}{k} - 3 \sum_{k=1}^n \frac{1}{k^2} + \frac{1}{n}}{n(n-1)} + 2 \frac{2 \sum_{k=1}^n \frac{1}{k} - 2 \sum_{k=1}^n \frac{1}{k^2} - 1 + \frac{1}{n}}{n(n-1)(n-2)} \right) \|x_j\|_2^2$$
$$- 2 \frac{\sum_{k=1}^n \frac{1}{k} - \sum_{k=1}^n \frac{1}{k^2} - \frac{1}{n} + \frac{1}{n^2}}{(n-1)(n-2)} \langle \sum_{i \in N} x_i, x_j \rangle + \frac{2 \sum_{k=1}^n \frac{1}{k} - 2 \sum_{k=1}^n \frac{1}{k^2} - 1 + \frac{1}{n}}{n(n-1)(n-2)} \| \sum_{i \in N} x_i \|_2^2$$
$$+ \left( \frac{\sum_{k=1}^n \frac{1}{k^2} - \frac{1}{n}}{n(n-1)} - \frac{2 \sum_{k=2}^n \frac{1}{k} - 2 \sum_{k=2}^n \frac{1}{k^2} - 1 + \frac{1}{n}}{n(n-1)(n-2)} \right) \left( \sum_{i \in N} \|x_i\|_2^2 \right)$$
$$+ 2 \frac{\sum_{k=1}^n \frac{1}{k} - \frac{1}{n}}{n-1} \langle x_j, \alpha \rangle - 2 \frac{\sum_{k=1}^n \frac{1}{k} - 1}{n(n-1)} \langle \sum_{i \in N} x_i, \alpha \rangle$$

$$\tag{4}$$

*Proof see Appendix B.*

The use of CHG Shapley to calculate the data value of each training data point is outlined in Algorithm 1, and the experimental results for different data valuation methods are presented in Table 1. Detailed experimental results comparing these data valuation methods can be found in Appendix A.

**Differences from Data Shapley**: Although our approach shares the same underlying principle as Data Shapley, we do not try to approximate it. Data Shapley utilizes a model trained on a subset of the training set, evaluated on an additional validation set, as its utility function. In CHG Shapley, our utility function captures the extent of training loss reduction when using a subset of the training dataset. Furthermore, in Data Shapley, the required size of the validation set can grow by orders of magnitude to achieve the desired utility precision (i.e., the model's accuracy on the validation set). For example, in a classification task, achieving a precision of 0.001 for the utility function may require 1,000 validation samples, while a precision of 0.0001 could demand 10,000 samples. Another challenge lies in the noise affecting the utility function, which arises from the model training process—particularly due to stochastic gradient descent—as discussed in Data Banzhaf (Wang & Jia, 2023). In contrast, CHG Shapley relies solely on the training set and incorporates both loss and gradient information from the model during training. This richer and more precise information allows CHG Shapley to achieve lower computational costs and higher precision. Moreover, by averaging the Shapley values over multiple epochs, CHG Shapley is somewhat resistant to the impacts of utility noise.

---

**Algorithm 1** CHG Shapley-based Data Valuation Algorithm

---

**Require:** Training dataset $N$, initial model parameters $\theta_0$, total epochs $K$
**Ensure:** Shapley value of each training data point in $N$
1: **for** $k$-th epoch **do**
2:     Acquire the loss and gradients for each data point in $N$, i.e., $h_i, \nabla f(i; \theta_k)$
3:     Using Equation 4 to calculate $\phi_i^k$, the Shapley value of $i$-th data point in $k$ epoch, where $\alpha = \frac{1}{N} \sum_{i \in N} h_i \nabla f(i; \theta)$ and $x_i = h_i \nabla f(i; \theta_k)$
4:     Train model on $N$, updating parameters to $\theta_{k+1}$
5: **end for**
6: Return the average Shapley value for each data point across epochs: $\sum_{k=1}^{K} \phi_i^k / K$

---

## 4 DATA SELECTION

### 4.1 FROM DATA VALUATION TO DATA SELECTION

The Shapley value is a widely used method for quantifying individual contributions to overall utility in cooperative games. Previous work (Ghorbani & Zou, 2019; Wang & Jia, 2023) has also heuristically prioritized data points with high Shapley values for model training. In this section, we aim to lay the theoretical groundwork for employing Shapley values in data selection.

**Theorem 2** *Let $N$ be a set of players and $U$ a utility function. The Shapley value of player $i \in N$ is denoted by $\phi_i(U)$. Suppose that for all $i$ and for all subsets $S \subseteq N \setminus \{i\}$, the utility difference satisfies $-m \leq U(S \cup \{i\}) - U(S) \leq M$. Then:*

*1. If $\phi_i(U) \geq 0$, then*

$$P_{\pi \sim \Pi}\left[ U(S_\pi^i \cup \{i\}) \geq U(S_\pi^i) \right] \geq \frac{\phi_i(U)}{M},$$

*2. If $\phi_i(U) \leq 0$, then*

$$P_{\pi \sim \Pi}\left[ U(S_\pi^i \cup \{i\}) \geq U(S_\pi^i) \right] \leq 1 + \frac{\phi_i(U)}{m}.$$

*Here, $S_\pi^i$ represents the set of players preceding player $i$ in a uniform random permutation $\pi$.*

*The proof can be found in Appendix C.*

The theorem offers a probabilistic interpretation of Shapley values. Supposing $S_\pi^i$ represents the dataset selected so far to some extent, then, if $i$ has a positive Shapley value, the probability that

it positively contributes to the coalition $S_\pi^i$ has a lower bound that increases proportionally with its Shapley value. Conversely, if $i$ has a negative Shapley value, the upper bound on this probability decreases as its Shapley value becomes smaller. Thus, when $i$ has a larger Shapley value, adding it to the selected training subset $S_\pi^i$ is more likely to yield a positive benefit for $S_\pi^i$. And when $S_\pi^i$ can not fairly approximate the dataset selected so far, using the original Shapley value for data selection may become ineffective.

---

**Algorithm 2** CHG Shapley-based Data Selection Algorithm

---

**Require:** Training dataset $N$, initial parameters $\theta_0$, total epochs $K$, selection interval $R = 20$, fraction $a$ of selected data.
**Ensure:** Final parameters $\theta_K$
 1: Initialize subset $S = \emptyset$.
 2: **for** $k$-th epoch **do**
 3:     **if** $k \mod R == 0$ **then**
 4:         $S = \emptyset$
 5:         **for** $c$-th class **do**
 6:             Acquire loss and last-layer gradients for $N_c$ (the subset of $N$ with label $c$), i.e., $h_i, \nabla f(i; \theta_k)$
 7:             Using Equation 4 to calculate Shapley values with $\alpha = \frac{1}{N_c} \sum_{i \in N_c} h_i \nabla f(i; \theta)$ and $x_i = h_i \nabla f(i; \theta_k)$
 8:             Add top $aN_c$ points with highest Shapley values to $S$
 9:         **end for**
10:     **end if**
11:     Train model on $S$, updating parameters to $\theta_{k+1}$
12: **end for**

---

## 4.2 CHG SHAPLEY-BASED DATA SELECTION ALGORITHM

In this section, we introduce several implementation strategies and practical techniques to improve the scalability and efficiency of CHG Shapley, incorporating common tricks from data selection methods (Killamsetty et al., 2021a;b), and all the compared methods (except Full, Random, and AdaptiveRandom) also adopt these tricks:

**Interval Data Selection**: Instead of selecting new data at every training step, we perform data selection at intervals. For example, during a 300-epoch training process, data selection is done every 20 epochs, while in the remaining epochs, the previously selected subset is used for training. This approach significantly reduces the computational overhead associated with data selection.

**Last-Layer Gradients**: Given the high dimensionality of gradients in modern deep learning models, we adopt a last-layer gradient approximation. By focusing solely on the gradients of the last layer, this technique accelerates the computation for all methods.

**Per-Class Approximations**: To further enhance model accuracy, we apply a per-class approach, running the algorithm separately for each class by considering only the data points from that class in each data selection iteration. This reduces both memory usage and computational cost, improving the overall performance of all methods.

We detail the CHG Shapley Data Selection Algorithm in Algorithm 2, integrating the aforementioned techniques to ensure efficient and scalable data valuation-based data selection.

## 5 EXPERIMENTS

We first present the data valuation experiments, followed by the data selection experiments. All experiments were conducted on a server equipped with an Intel(R) Xeon(R) Gold 6326 CPU (2.90 GHz) and an NVIDIA A40 GPU. All methods were implemented using PyTorch.

Table 2: The accuracy (%) and time (h) comparison. All methods are trained with ResNet-18.

| Dataset | CIFAR10 | | | | | CIFAR100 | | | | |
|---|---|---|---|---|---|---|---|---|---|---|
| Fraction | 0.05 | 0.1 | 0.3 | 0.5 | 0.7 | 0.05 | 0.1 | 0.3 | 0.5 | 0.7 |
| **Accuracy (%)** | | | | | | | | | | |
| Full | | | 95.51 | | | | | 77.56 | | |
| Random | 66.63 | 77.36 | 89.93 | 93.11 | 93.87 | 22.58 | 36.51 | 62.27 | 69.70 | 73.04 |
| AdaptiveRandom | 83.26 | 88.78 | 93.88 | 94.58 | 94.66 | 48.52 | 60.58 | 72.45 | 74.97 | 75.70 |
| Glister | 85.42 | 90.52 | 89.80 | 90.15 | 93.36 | 46.54 | **63.88** | 71.88 | 73.93 | 75.25 |
| GradMatchPB | 84.91 | 90.14 | 93.73 | 94.44 | 94.54 | 49.01 | 62.56 | 71.27 | 73.66 | 74.43 |
| Hardness Shapley | 63.07 | 79.96 | 93.93 | 94.63 | 94.78 | 37.78 | 55.21 | **72.46** | **75.86** | 75.77 |
| TracIn (Gradient-Dot) | 63.47 | 81.49 | **94.13** | **94.85** | 94.62 | 40.21 | 56.55 | 72.11 | 75.01 | **76.33** |
| CGSV (Gradient-Cosine) | 72.97 | 83.49 | 92.24 | 93.68 | 93.98 | 44.08 | 53.87 | 68.16 | 74.04 | 75.59 |
| GradE Shapley (This paper) | 83.10 | 89.80 | 93.10 | 93.81 | 93.58 | 50.38 | 60.16 | 70.53 | 73.66 | 74.57 |
| CHG Shapley (This paper) | **85.47** | **91.20** | 93.73 | 94.42 | 94.07 | **53.20** | 63.70 | 72.32 | 74.59 | 75.31 |
| **Time (h)** | | | | | | | | | | |
| Full | | | 2.2 | | | | | 2.2 | | |
| Random | 0.1 | 0.3 | 0.5 | 1.8 | 2.4 | 0.2 | 0.5 | 1.1 | 0.9 | 1.6 |
| AdaptiveRandom | 0.1 | 0.4 | 1.1 | 1.5 | 1.8 | 0.2 | 0.4 | 0.9 | 1.3 | 2.0 |
| Glister | 0.4 | 0.8 | 2.2 | 2.4 | 2.3 | 0.4 | 1.0 | 3.2 | 3.2 | 3.6 |
| GradMatchPB | 0.2 | 0.4 | 1.0 | 1.4 | 2.3 | 0.3 | 0.5 | 1.2 | 2.2 | 3.0 |
| Hardness Shapley | 0.3 | 0.6 | 1.0 | 1.8 | 2.4 | 0.4 | 0.8 | 1.3 | 2.9 | 3.6 |
| TracIn (Gradient-Dot) | 0.3 | 0.5 | 1.2 | 1.8 | 2.4 | 0.4 | 0.6 | 1.3 | 2.9 | 2.4 |
| CGSV (Gradient-Cosine) | 0.4 | 0.5 | 1.1 | 1.7 | 2.4 | 0.4 | 0.8 | 1.2 | 2.8 | 2.4 |
| GradE Shapley (This paper) | 0.3 | 0.6 | 1.2 | 1.8 | 2.7 | 0.4 | 0.5 | 1.3 | 1.9 | 3.6 |
| CHG Shapley (This paper) | 0.3 | 0.6 | 1.4 | 1.7 | 2.2 | 0.4 | 0.5 | 1.3 | 2.1 | 3.5 |

## 5.1 DATA VALUATION SETUP AND RESULTS

Our CHG Shapley data valuation method (Algorithm 1) was implemented using the OpenDataVal framework (Jiang et al., 2023), and experiments were conducted on the CIFAR-10 embedding dataset. The model used for evaluation was a pre-trained ResNet50 followed by logistic regression, with results reported in Table 1.

The methods compared include Leave-One-Out, Influence Subsample (an efficient approximation of the Influence function (Koh & Liang, 2017)), DVRL (Yoon et al., 2020), Data Banzhaf (Wang & Jia, 2023), AME (Lin et al., 2022), LAVA (Just et al., 2023), and Data OOB (Kwon & Zou, 2023). We do not include Data Shapley and Bata Shapley because these two methods are too slow. We evaluated performance on three tasks: noisy label detection (Table 5), noisy feature detection (Fig. 2), and point removal experiments (Fig. 3). Detailed results comparing these methods can be found in Appendix A.

Across the three tasks, Data OOB and CHG Shapley ranked first and second, respectively. Notably, CHG Shapley completed its evaluation in 18 seconds, whereas Data OOB took over 2 hours. These results demonstrate that CHG Shapley not only achieves competitive data valuation and noise detection performance but also offers significant operational efficiency. This efficiency is driven by the analytic form of the proposed utility function, which evaluates the influence of data subsets on training loss. In the next section, we explore CHG Shapley's effectiveness in data selection for larger, modern datasets.

## 5.2 DATA SELECTION SETUP

In this experiment, we employed a ResNet-18 model, trained from scratch using stochastic gradient descent with a learning rate of 0.05, momentum of 0.9, weight decay set to 5e-4, and Nesterov acceleration. The model was trained over 300 epochs, with data reselection occurring every 20 epochs. The fraction of selected data varied across 0.05, 0.1, 0.3, 0.5, and 0.7.

In the class imbalance experiments, we set the imbalance ratio to 0.3, meaning that for 30% of the classes, only 10% of their training and validation data was retained. In the noisy label experiments, we set the noise ratio to 0.3, meaning that 30% of the labels in the training set were randomly replaced with labels from the available classes.

Table 3: Comparison of performance under class imbalance setting.

| Dataset | CIFAR10 | | | | | CIFAR100 | | | | |
|---|---|---|---|---|---|---|---|---|---|---|
| Fraction | 0.05 | 0.1 | 0.3 | 0.5 | 0.7 | 0.05 | 0.1 | 0.3 | 0.5 | 0.7 |
| Full | | | 90.37 | | | | | 65.11 | | |
| AdaptiveRandom | 73.64 | 82.00 | **88.73** | 89.61 | **89.92** | 36.45 | 49.44 | 61.38 | 63.44 | 64.27 |
| Glister | 67.80 | 81.47 | 83.16 | 82.07 | 88.63 | 32.79 | 48.15 | 60.86 | 61.94 | 62.72 |
| GradMatchPB | **75.12** | 82.92 | 88.21 | 89.55 | 89.71 | 39.45 | 51.39 | **61.54** | **63.97** | 64.07 |
| Hardness Shapley | 35.57 | 64.95 | 87.95 | 89.13 | 89.56 | 25.87 | 43.34 | 61.42 | 63.43 | 64.31 |
| TracIn (Gradient-Dot) | 42.97 | 68.34 | 88.45 | 89.42 | 89.54 | 31.81 | 43.52 | 61.10 | 63.54 | 64.02 |
| CGSV (Gradient-Cosine) | 63.09 | 74.45 | 87.60 | **89.72** | 89.01 | 32.35 | 42.58 | 58.03 | 63.0 | **64.38** |
| GradE Shapley (This paper) | 67.61 | 80.98 | 86.46 | 87.55 | 88.55 | 39.65 | 48.96 | 57.81 | 61.02 | 62.14 |
| CHG Shapley (This paper) | 69.94 | **83.65** | 87.69 | 88.61 | 88.86 | **40.31** | **51.88** | 60.33 | 62.90 | 63.89 |

Table 4: Comparison of performance under noisy label setting.

| Dataset | CIFAR10 | | | | | CIFAR100 | | | | |
|---|---|---|---|---|---|---|---|---|---|---|
| Fraction | 0.05 | 0.1 | 0.3 | 0.5 | 0.7 | 0.05 | 0.1 | 0.3 | 0.5 | 0.7 |
| Full | | | 79.95 | | | | | 56.67 | | |
| AdaptiveRandom | 57.18 | 63.21 | 71.84 | 75.38 | 77.41 | 24.12 | 32.69 | 45.36 | 52.18 | 53.00 |
| Glister | 62.48 | 71.22 | 75.51 | 77.62 | 77.91 | 18.75 | 27.23 | 46.22 | 49.38 | 52.93 |
| GradMatchPB | 56.33 | 69.29 | 73.50 | 76.44 | 76.74 | 26.40 | 32.55 | 43.75 | 48.47 | 50.50 |
| Hardness Shapley | 21.63 | 35.41 | 63.31 | 75.95 | 77.28 | 3.72 | 7.23 | 43.16 | 50.90 | 51.80 |
| TracIn (Gradient-Dot) | 30.72 | 46.87 | 70.45 | 77.14 | 77.90 | 28.37 | 29.70 | 43.05 | 50.83 | 52.29 |
| CGSV (Gradient-Cosine) | 70.85 | 77.48 | 78.90 | 78.42 | 76.84 | 33.12 | 42.12 | 55.31 | 59.23 | 57.18 |
| CHG Shapley (This paper) | 72.18 | 83.35 | 88.57 | 90.00 | 90.20 | 36.75 | 46.93 | 60.09 | 62.59 | 62.79 |
| GradE Shapley (This paper) | **79.35** | **86.74** | **90.83** | **92.03** | **91.35** | **44.04** | **52.14** | **60.25** | **63.20** | **62.92** |

The compared methods include Full (training the model with the full dataset), Random (selecting data in the first epoch and never updating the selection), AdaptiveRandom (selecting data randomly in every selection epoch), Glister (Killamsetty et al., 2021b) and GradMatchPB (Killamsetty et al., 2021a) (two coreset selection methods), Hardness Shapley (selecting data with the highest loss), TracIn (Pruthi et al., 2020), and CGSV (Xu et al., 2021) (which use gradient inner product or cosine similarity for data selection as in (Xia et al., 2024)), as well as our proposed GradE Shapley and CHG Shapley. To ensure a fair comparison, the target gradient for TracIn, CGSV, GradE Shapley, and CHG Shapley is the mean gradient of all training data, eliminating the need for an additional validation set. All methods were implemented using the Cords framework (Killamsetty et al., 2021b;a).

## 5.3 DATA SELECTION RESULTS

We present the accuracy and time consumption comparisons in Table 2 under the standard dataset setting. Hardness Shapley also performs competitively at higher fractions (e.g., 0.3), aligning with the findings of Qin et al. (2024). However, when the selection ratio is small (e.g., 0.05), its performance significantly declines, likely due to the difficulty in training the network when all selected data points are particularly hard. TracIn also performs well when the selection ratio is large. CHG Shapley demonstrates particularly strong performance when the selection ratio is small (i.e., 0.05, 0.1), effectively identifying high-value data. Furthermore, its time consumption is comparable to other methods, underscoring the efficiency of CHG Shapley as a data valuation approach.

Table 3 presents the results under the class imbalance dataset setting. Similar to the standard dataset setting, CHG Shapley performs well at smaller selection ratios, and its comparison with GradE Shapley highlights the advantage of incorporating data hardness as part of data value.

The most surprising results are presented in Table 4, under the noisy label dataset setting. GradE Shapley demonstrates a 5%-10% performance improvement over other methods (except CHG Shapley) in this scenario. Even with 10%-30% of the data selected, GradE Shapley can outperform using the full dataset, reinforcing its effectiveness in identifying label noise. In contrast, Hardness Shapley performs notably worse in this setting, especially at smaller selection ratios. In other words, the unreliability of hardness in the presence of label noise leads to CHG Shapley underperforming relative to GradE Shapley.

Overall, the combination of efficiency and accuracy demonstrated across these three experiments establishes CHG Shapley as a practical and powerful tool for data selection and valuation in large-scale datasets with small amounts of label noise. And GradE Shapley excels when handling larger amounts of label noise.

## 6 DISCUSSION AND FUTURE WORK

In the data selection experiments, the introduction of hardness improves the model's performance when no label noise is present. This suggests that hardness can serve as a proxy for data value to some extent. However, in cases with significant label noise, incorporating hardness into GradE Shapley diminishes its effectiveness. This indicates a need for further investigation into how hardness can be fairly integrated into data value assessments, particularly when it becomes an unreliable metric.

Besides, CHG Shapley performs well at small fractions but struggles with larger ones. This challenge may arise from the difficulties of selecting data based on Shapley values (Wang et al., 2024c), rather than the limitations of CHG Shapley itself. Specifically, once a subset of data $s$ is selected, the Shapley value of the remaining data may change, as the set $S_\pi^i$ in Equation 2 will include $s$. This affects the effectiveness of the original Shapley values in evaluating the remaining data. If we must recalculate the Shapley values for the remaining data each time a new data point is selected, the time complexity becomes unmanage able even with our proposed method. Thus, we leave the exploration of iterative Shapley value calculations for data selection to future work.

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

# A   DATA VALUATION RESULTS

We adhere to the experimental settings and implementation described in (Jiang et al., 2023) to conduct the data valuation experiment. We evaluate several data valuation methods on the CIFAR10-embeddings dataset, using a pretrained ResNet50 model for feature extraction, followed by a logistic regression model for classification. The dataset contains 1,000 training samples, 100 validation samples, and 300 test samples. The noise rate is set as 0.1, which means 10% of the labels in both the training and validation datasets will be randomly altered.

We assess the performance using classification accuracy. The logistic regression model is trained for 10 epochs with a batch size of 100 and a learning rate of 0.01. The data valuation methods compared in this study include CHG Shapley, Leave-One-Out, Influence Subsample (an efficient approximation of the Influence function), DVRL, Data Banzhaf, AME, LAVA, and Data OOB. We have excluded the results for Data Shapley and Beta Shapley, as these two methods are at least 10 times more time-consuming than the methods evaluated. The runtime comparison results are provided in Table 6.

## A.1   NOISY LABEL DETECTION

We conduct noisy data detection by introducing mislabeled data into the dataset and applying various data valuation algorithms to identify the mislabeled instances. The performance of each algorithm is measured using the F1-score, which evaluates the balance between precision and recall. A higher F1-score indicates better accuracy in detecting mislabeled data.

As shown in Table 5, Data OOB achieves the highest F1-score, while CHG Shapley follows closely as the second-best performer. This indicates that CHG Shapley demonstrates competitive noisy data detection capabilities compared to state-of-the-art data valuation methods.

Table 5: F1-scores of various data valuation methods for noisy data detection

| Method | F1-Score |
|---|---|
| Leave-One-Out | 0.184573 |
| Random Evaluator | 0.173623 |
| AME (1000 models) | 0.181989 |
| DVRL (2000 rl epochs) | 0.297872 |
| Data Banzhaf (1000 models) | 0.160883 |
| Data OOB (1000 models) | **0.368821** |
| Influence Subsample (1000 models) | 0.199662 |
| Lava Evaluator | 0.212329 |
| CHG Shapley | 0.334405 |

## A.2   NOISY FEATURE DETECTION

In Fig. 2, we visualize the effectiveness of each data valuation method in identifying noisy data points by inspecting a fraction of the dataset. The x-axis represents the proportion of inspected data, while the y-axis indicates the proportion of identified corrupted data. The orange curve represents the optimal performance, while the blue curve reflects the true performance of the evaluator. A closer alignment of the blue curve with the orange curve indicates a more effective data valuation method.

Notably, CHG Shapley and Data OOB emerge as the top performers, demonstrating their superior ability to detect noisy samples compared to other evaluated methods.

## A.3   POINT REMOVAL EXPERIMENT

In this experiment, we evaluate the impact of removing data points based on the rankings generated by each data valuation algorithm. After each removal of high- or low-value data points, we retrain the model and assess its performance on a test dataset. The x-axis represents the fraction of removed high or low-value data, while the y-axis indicates the accuracy of the retrained model using the

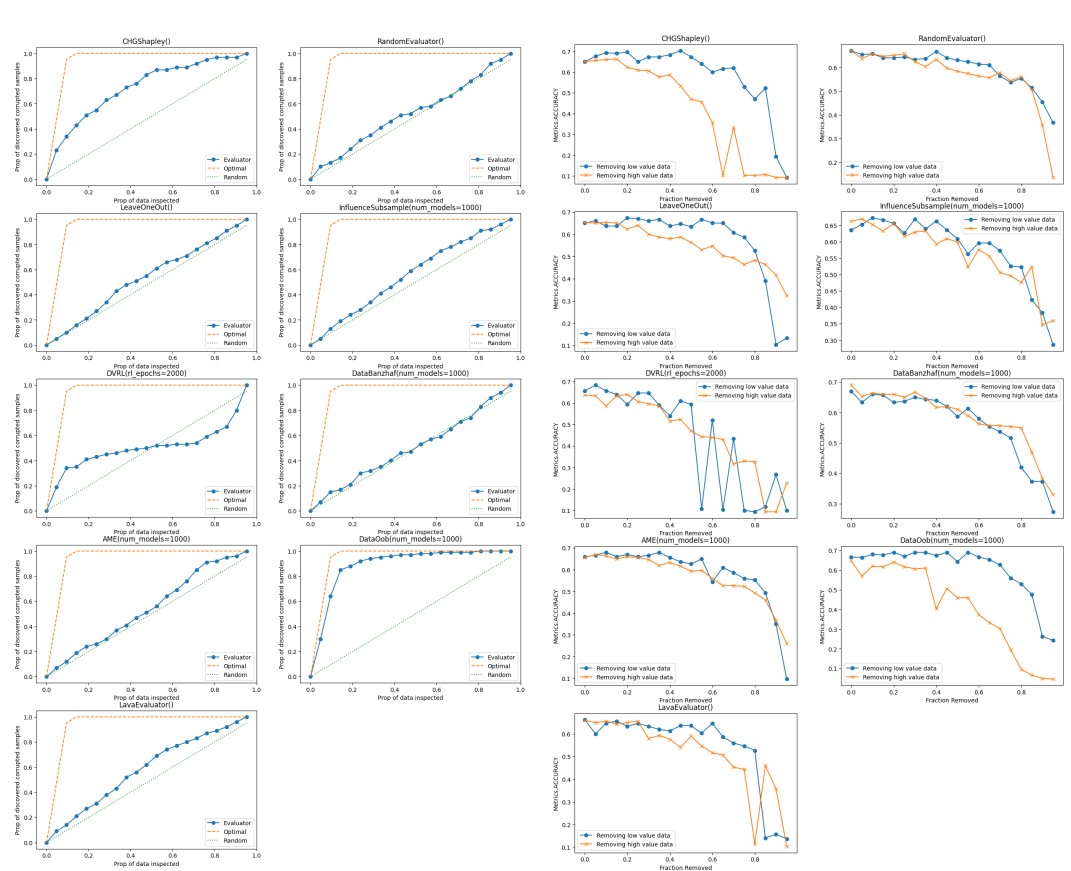

Figure 2: Noisy feature detection experiment on CIFAR10-embeddings dataset.

Figure 3: Point removal experiment on CIFAR10-embeddings dataset.

remaining data. Two curves are generated: one illustrating performance when data is removed in descending order (orange) and the other in ascending order (blue). For the orange curve, a lower accuracy indicates better ability to detect high-value data, while a higher accuracy is preferable for the blue curve cause removing low-value data should have little impact to model accuracy.

As shown in Fig. 3, both CHG Shapley and Data OOB perform exceptionally well in this experiment, reinforcing their positions as leading methods for effective data valuation and highlighting their capacity to optimize the removal of noisy samples for enhanced model performance.

## A.4 RUNTIME COMPARISON

The runtime comparison in Table 6 shows that CHG Shapley is one of the fastest methods, completing in just 18 seconds, while methods like Leave-One-Out and Data OOB take over an hour. Data Shapley and Beta Shapley, on the other hand, are far slower, requiring more than 24 hours. This highlights CHG Shapley's efficiency for practical use.

Table 6: Comparison of runtimes for different data valuation methods on CIFAR10-embeddings

| Method | Elapsed Time (hh:mm:ss) |
|---|---|
| Leave-One-Out | 1:41:25 |
| Influence Subsample (1000 models) | 1:11:42 |
| DVRL(2000 rl epochs) | 00:06:10 |
| Data Banzhaf (1000 models) | 1:04:06 |
| AME (1000 models) | 4:38:19 |
| Data OOB (1000 models) | 2:02:03 |
| Data Shapley (1000 models) | More than 24h |
| Beta Shapley (1000 models) | More than 24h |
| Lava Evaluator | **00:00:02** |
| CHG Shapley | 00:00:18 |

## B   PROOF OF THEOREM 1

Because $U(S) = ||\alpha||^2 - ||\frac{1}{|S|} \sum_{i \in S} x_i - \alpha||^2 = -||\frac{1}{|S|} \sum_{i \in S} x_i||^2 + 2\frac{1}{|S|} \sum_{i \in S} \langle x_i, \alpha \rangle$, and we define $U_1(S) = -||\frac{1}{|S|} \sum_{i \in S} x_i||^2$ and $U_2(S) = 2\frac{1}{|S|} \sum_{i \in S} x_i \alpha$, then $U(S) = U_1(S) + 2U_2(S)$.

The basic idea here is to use the Linearity property of the Shapley value. By calculating $\phi_j(U_1)$ and $\phi_j(U_2)$ separately, we can get $j$-th player's Shapley value as: $\phi_j(U) = \phi_j(U_1) + 2\phi_j(U_2)$.

For $U_2$, we have:

$$U_2(S \cup j) - U_2(S) = \frac{1}{|S|+1} \sum_{i \in S \cup j} \langle x_i, \alpha \rangle - \frac{1}{|S|} \sum_{i \in S} \langle x_i, \alpha \rangle$$

$$= \frac{\langle x_j, \alpha \rangle}{|S|+1} - \frac{\sum_{i \in S} \langle x_i, \alpha \rangle}{|S|(|S|+1)}.$$

Then we have:

$$\phi_j(U_2) = \frac{1}{n} \sum_{k=1}^{n} \binom{n-1}{k-1}^{-1} \sum_{S \subseteq N \setminus \{j\}, |S|=k-1} [U_2(S \cup j) - U_2(S)]$$

$$= \frac{1}{n} \sum_{k=1}^{n} \binom{n-1}{k-1}^{-1} \sum_{\substack{S \subseteq N \setminus \{j\}, \\ |S|=k-1}} \frac{\langle x_j, \alpha \rangle}{|S|+1} - \frac{1}{n} \sum_{k=1}^{n} \binom{n-1}{k-1}^{-1} \sum_{\substack{S \subseteq N \setminus \{j\}, \\ |S|=k-1}} \frac{\sum_{i \in S} \langle x_i, \alpha \rangle}{|S|(|S|+1)}$$

$$= \frac{1}{n} \sum_{k=1}^{n} \binom{n-1}{k-1}^{-1} \binom{n-1}{k-1} \frac{\langle x_j, \alpha \rangle}{k} - \frac{1}{n} \sum_{k=2}^{n} \binom{n-1}{k-1}^{-1} \sum_{i \in N \setminus \{j\}} \sum_{\substack{S \subseteq N \setminus \{i,j\}, \\ |S|=k-2}} \frac{\langle x_i, \alpha \rangle}{(k-1)k}$$

$$= \frac{1}{n} \sum_{k=1}^{n} \frac{\langle x_j, \alpha \rangle}{k} - \frac{1}{n} \sum_{k=2}^{n} \binom{n-1}{k-1}^{-1} \sum_{i \in N \setminus \{j\}} \binom{n-2}{k-2} \frac{\langle x_i, \alpha \rangle}{(k-1)k}$$

$$= \frac{\sum_{k=1}^{n} \frac{1}{k}}{n} \langle x_j, \alpha \rangle - \frac{1}{n} \sum_{k=2}^{n} \frac{k-1}{n-1} \sum_{i \in N \setminus \{j\}} \frac{\langle x_i, \alpha \rangle}{(k-1)k}$$

$$= \frac{\sum_{k=1}^{n} \frac{1}{k}}{n} \langle x_j, \alpha \rangle - \frac{\sum_{k=2}^{n} \frac{1}{k}}{n(n-1)} \langle \sum_{i \in N \setminus \{j\}} x_i, \alpha \rangle$$

$$= \frac{\sum_{k=1}^{n} \frac{1}{k} - \frac{1}{n}}{n-1} \langle x_j, \alpha \rangle - \frac{\sum_{k=1}^{n} \frac{1}{k} - 1}{n(n-1)} \langle \sum_{i \in N} x_i, \alpha \rangle.$$

Thus, $\phi_j(U_2) = \frac{\sum_{k=1}^{n} \frac{1}{k} - \frac{1}{n}}{n-1} \langle x_j, \alpha \rangle - \frac{\sum_{k=1}^{n} \frac{1}{k} - 1}{n(n-1)} \langle \sum_{i \in N} x_i, \alpha \rangle$.

Similarily, for $U_1$, we have:

$$U_1(S \cup j) - U_1(S) = -\frac{1}{(|S|+1)^2} x_j^2 - \frac{2}{(|S|+1)^2} x_j (\sum_{i \in S} x_i) + \frac{2|S|+1}{|S|^2(|S|+1)^2} (\sum_{i \in S} x_i)^2.$$

Then we have:

$$\phi_j(U_1) = -\left( \frac{1}{n} \sum_{k=1}^{n} \frac{1}{k^2} \right) x_j^2 - \left( \frac{2}{n(n-1)} \sum_{k=2}^{n} \frac{(k-1)}{k^2} \right) (\sum_{i \in N/j} x_i) x_j$$

$$+ \left( \frac{1}{n(n-1)} \sum_{k=2}^{n} \frac{(2k-1)}{k^2(k-1)} \right) (\sum_{i \in N/j} x_i^2)$$

$$+ \left( \frac{1}{n(n-1)(n-2)} \sum_{k=3}^{n} \frac{(2k-1)(k-2)}{k^2(k-1)} \right) (\sum_{a \in N/j} \sum_{b \in N/j, a \neq b} x_a x_b)$$

Then:

$$\phi_j(U_1) = - \left( \frac{1}{n} \sum_{k=1}^{n} \frac{1}{k^2} \right) x_j^2 + \left( \frac{2}{n(n-1)} \sum_{k=2}^{n} \frac{(k-1)}{k^2} \right) x_j^2 - \left( \frac{2}{n(n-1)} \sum_{k=2}^{n} \frac{(k-1)}{k^2} \right) (\sum_{i \in N} x_i) x_j$$

$$+ \left( \frac{1}{n(n-1)} \sum_{k=2}^{n} \frac{(2k-1)}{k^2(k-1)} \right) (\sum_{i \in N} x_i^2) - \left( \frac{1}{n(n-1)} \sum_{k=2}^{n} \frac{(2k-1)}{k^2(k-1)} \right) x_j^2$$

$$- \left( \frac{1}{n(n-1)(n-2)} \sum_{k=3}^{n} \frac{(2k-1)(k-2)}{k^2(k-1)} \right) (\sum_{i \in N} x_i^2)$$

$$+ \left( \frac{1}{n(n-1)(n-2)} \sum_{k=3}^{n} \frac{(2k-1)(k-2)}{k^2(k-1)} \right) (\sum_{a \in N} \sum_{b \in N} x_a x_b)$$

$$- 2 \left( \frac{1}{n(n-1)(n-2)} \sum_{k=3}^{n} \frac{(2k-1)(k-2)}{k^2(k-1)} \right) (\sum_{a \in N} x_a x_j)$$

$$+ 2 \left( \frac{1}{n(n-1)(n-2)} \sum_{k=3}^{n} \frac{(2k-1)(k-2)}{k^2(k-1)} \right) x_j^2$$

Then we have:

$$\phi_j(U_1)$$

$$= [-\frac{1}{n} \sum_{k=1}^{n} \frac{1}{k^2} + \frac{2}{n(n-1)} \sum_{k=2}^{n} \frac{(k-1)}{k^2} - \frac{1}{n(n-1)} \sum_{k=2}^{n} \frac{(2k-1)}{k^2(k-1)} + \frac{2}{n(n-1)(n-2)} \sum_{k=3}^{n} \frac{(2k-1)(k-2)}{k^2(k-1)}] x_j^2$$

$$- \left( \frac{2}{n(n-1)} \sum_{k=2}^{n} \frac{(k-1)}{k^2} \right) (\sum_{i \in N} x_i) x_j - 2 \left( \frac{1}{n(n-1)(n-2)} \sum_{k=3}^{n} \frac{(2k-1)(k-2)}{k^2(k-1)} \right) (\sum_{i \in N} x_i) x_j$$

$$+ \left( \frac{1}{n(n-1)(n-2)} \sum_{k=3}^{n} \frac{(2k-1)(k-2)}{k^2(k-1)} \right) (\sum_{a \in N} \sum_{b \in N} x_a x_b)$$

$$+ \left( \frac{1}{n(n-1)} \sum_{k=2}^{n} \frac{(2k-1)}{k^2(k-1)} - \frac{1}{n(n-1)(n-2)} \sum_{k=3}^{n} \frac{(2k-1)(k-2)}{k^2(k-1)} \right) (\sum_{i \in N} x_i^2),$$

And:

$$\phi_j(U_1)$$

$$= [-\frac{1}{n} \sum_{k=1}^{n} \frac{1}{k^2} + \frac{1}{n(n-1)} \left( 2 \sum_{k=1}^{n} \frac{1}{k} - 3 \sum_{k=1}^{n} \frac{1}{k^2} + \frac{1}{n} \right) + 2 \frac{2 \sum_{k=2}^{n} \frac{1}{k} - 2 \sum_{k=2}^{n} \frac{1}{k^2} - 1 + \frac{1}{n}}{n(n-1)(n-2)}] x_j^2$$

$$- \frac{2}{n(n-1)} \left( \sum_{k=2}^{n} \frac{1}{k} - \sum_{k=2}^{n} \frac{1}{k^2} + \frac{1}{(n-2)} \left( 2 \sum_{k=2}^{n} \frac{1}{k} - 2 \sum_{k=2}^{n} \frac{1}{k^2} - 1 + \frac{1}{n} \right) \right) (\sum_{i \in N} x_i) x_j$$

$$+ \frac{1}{n(n-1)(n-2)} \left( 2 \sum_{k=2}^{n} \frac{1}{k} - 2 \sum_{k=2}^{n} \frac{1}{k^2} - 1 + \frac{1}{n} \right) (\sum_{a \in N} \sum_{b \in N} x_a x_b)$$

$$+ \left( \frac{1}{n(n-1)} \left( \sum_{k=2}^{n} \frac{1}{k^2} + 1 - \frac{1}{n} \right) - \frac{2 \sum_{k=2}^{n} \frac{1}{k} - 2 \sum_{k=2}^{n} \frac{1}{k^2} - 1 + \frac{1}{n}}{n(n-1)(n-2)} \right) (\sum_{i \in N} x_i^2),$$

Thus:

$$\phi_j(U_1)$$

$$= [-\frac{1}{n}\sum_{k=1}^{n}\frac{1}{k^2} + \frac{1}{n(n-1)}\left(2\sum_{k=1}^{n}\frac{1}{k} - 3\sum_{k=1}^{n}\frac{1}{k^2} + \frac{1}{n}\right) + 2\frac{2\sum_{k=1}^{n}\frac{1}{k} - 2\sum_{k=1}^{n}\frac{1}{k^2} - 1 + \frac{1}{n}}{n(n-1)(n-2)}]x_j^2$$

$$-\frac{2}{(n-1)(n-2)}\left(\sum_{k=1}^{n}\frac{1}{k} - \sum_{k=1}^{n}\frac{1}{k^2} - \frac{(n-1)}{n^2}\right)(\sum_{i\in N}x_i)x_j$$

$$+\frac{1}{n(n-1)(n-2)}\left(2\sum_{k=1}^{n}\frac{1}{k} - 2\sum_{k=1}^{n}\frac{1}{k^2} - 1 + \frac{1}{n}\right)(\sum_{a\in N}\sum_{b\in N}x_a x_b)$$

$$+\left(\frac{1}{n(n-1)}\left(\sum_{k=1}^{n}\frac{1}{k^2} - \frac{1}{n}\right) - \frac{2\sum_{k=2}^{n}\frac{1}{k} - 2\sum_{k=2}^{n}\frac{1}{k^2} - 1 + \frac{1}{n}}{n(n-1)(n-2)}\right)(\sum_{i\in N}x_i^2),$$

Finally, we get the $j$-th player's Shapley value:

$$\phi_j(U)$$

$$= [-\frac{1}{n}\sum_{k=1}^{n}\frac{1}{k^2} + \frac{1}{n(n-1)}\left(2\sum_{k=1}^{n}\frac{1}{k} - 3\sum_{k=1}^{n}\frac{1}{k^2} + \frac{1}{n}\right) + 2\frac{2\sum_{k=1}^{n}\frac{1}{k} - 2\sum_{k=1}^{n}\frac{1}{k^2} - 1 + \frac{1}{n}}{n(n-1)(n-2)}]x_j^2$$

$$-\frac{2}{(n-1)(n-2)}\left(\sum_{k=1}^{n}\frac{1}{k} - \sum_{k=1}^{n}\frac{1}{k^2} - \frac{1}{n} + \frac{1}{n^2}\right)(\sum_{i\in N}x_i)x_j$$

$$+\frac{1}{n(n-1)(n-2)}\left(2\sum_{k=1}^{n}\frac{1}{k} - 2\sum_{k=1}^{n}\frac{1}{k^2} - 1 + \frac{1}{n}\right)(\sum_{a\in N}\sum_{b\in N}x_a x_b)$$

$$+\left(\frac{1}{n(n-1)}\left(\sum_{k=1}^{n}\frac{1}{k^2} - \frac{1}{n}\right) - \frac{2\sum_{k=2}^{n}\frac{1}{k} - 2\sum_{k=2}^{n}\frac{1}{k^2} - 1 + \frac{1}{n}}{n(n-1)(n-2)}\right)(\sum_{i\in N}x_i^2)$$

$$+\frac{2}{n-1}\left(\sum_{k=1}^{n}\frac{1}{k} - \frac{1}{n}\right)x_j\alpha - \frac{2}{n(n-1)}\left(\sum_{k=1}^{n}\frac{1}{k} - 1\right)(\sum_{i\in N}x_i)\alpha$$

## C  PROOF OF THEOREM 2

**Theorem 2**: *Let $N$ be a set of players and $U$ a utility function. The Shapley value of player $i \in N$ is denoted by $\phi_i(U)$. Suppose that for all $i$ and for all subsets $S \subseteq N \setminus \{i\}$, the utility difference satisfies $-m \leq U(S \cup \{i\}) - U(S) \leq M$. Then:*

1. *If $\phi_i(U) \geq 0$, then*

$$P_{\pi \sim \Pi}\left[U(S_\pi^i \cup \{i\}) \geq U(S_\pi^i)\right] \geq \frac{\phi_i(U)}{M},$$

2. *If $\phi_i(U) \leq 0$, then*

$$P_{\pi \sim \Pi}\left[U(S_\pi^i \cup \{i\}) \geq U(S_\pi^i)\right] \leq 1 + \frac{\phi_i(U)}{m}.$$

*Here, $S_\pi^i$ represents the set of players preceding player $i$ in a uniform random permutation $\pi$.*

Our proof strategy is first establishing a variant of Markov's inequality, then substituting $U(S_\pi^i \cup \{i\}) - U(S_\pi^i)$ as the variable into the inequality.

### C.1  A VARIANT OF MARKOV'S INEQUALITY

1. For $\mathbb{E}[X] \geq 0$:

   Since $X \cdot I[X < 0] \leq 0$, we have $\mathbb{E}[X \cdot I[X < 0]] \leq 0$. Therefore, we can state:

   $$\mathbb{E}[X] \leq \mathbb{E}[X \cdot I[X \geq 0]].$$

   Given that $X \leq M$, we obtain:

   $$X \cdot I[X \geq 0] \leq M \cdot I[X \geq 0].$$

   Thus,

   $$\mathbb{E}[X \cdot I[X \geq 0]] \leq \mathbb{E}[M \cdot I[X \geq 0]] = M \cdot P[X \geq 0].$$

   This leads us to:

   $$\mathbb{E}[X] \leq M \cdot P[X \geq 0].$$

   Rearranging gives:

   $$P[X \geq 0] \geq \frac{\mathbb{E}[X]}{M}.$$

2. For $\mathbb{E}[X] \leq 0$:

   Similarly, since $X \cdot I[X \geq 0] \geq 0$, it follows that $\mathbb{E}[X \cdot I[X \geq 0]] \geq 0$. Therefore, we have:

   $$\mathbb{E}[X] \geq \mathbb{E}[X \cdot I[X < 0]].$$

   Since $X \geq -m$, we find:

   $$X \cdot I[X < 0] \geq -m \cdot I[X < 0].$$

   Thus,

   $$\mathbb{E}[X \cdot I[X < 0]] \geq \mathbb{E}[-m \cdot I[X < 0]] = -m \cdot P[X < 0].$$

   This allows us to write:

   $$\mathbb{E}[X] \geq -m \cdot P[X < 0].$$

   Rearranging leads to:

   $$P[X < 0] \leq \frac{\mathbb{E}[X]}{-m}.$$

   Consequently, we have:

   $$P[X \geq 0] \geq 1 + \frac{\mathbb{E}[X]}{m}.$$

## C.2 PROOF

Let $X := U(S_\pi^i \cup \{i\}) - U(S_\pi^i)$, then the expectation of $X$ over the uniformly random permutation $\pi$ is:

$$\mathbb{E}_{\pi \sim \Pi} \left[ U(S_\pi^i \cup \{i\}) - U(S_\pi^i) \right] = \phi_i(U).$$

1. If $\phi_i(U) \geq 0$, by applying the first inequality in C.1: $P[X \geq 0] \geq \frac{\mathbb{E}[X]}{M}$. , we get:

$$P_{\pi \sim \Pi} \left[ U(S_\pi^i \cup \{i\}) \geq U(S_\pi^i) \right] \geq \frac{\phi_i(U)}{M}.$$

2. If $\phi_{\text{shap}}(i; U) \leq 0$, applying the second inequality in C.1: $P[X \geq 0] \leq 1 + \frac{\mathbb{E}[X]}{m}$. , we get:

$$P_{\pi \sim \Pi} \left[ U(S_\pi^i \cup \{i\}) \geq U(S_\pi^i) \right] \leq 1 + \frac{\phi_i(U)}{m}.$$

