# OpenReview forum: "CHG Shapley: Efficient Data Valuation and Selection towards Trustworthy Machine Learning"
_ICLR.cc/2025/Conference — Submitted to ICLR 2025_

### Official Review · Reviewer_jhq7 · 2024-11-01

**Soundness:** 2
**Presentation:** 2
**Contribution:** 2
**Rating:** 5
**Confidence:** 4

**Summary:**

The paper studies the problem of data valuation and an application of data valuation, namely data selection during training. The authors focus on the efficiency of data valuation for large datasets, which presents an issue for many common Shapley value-based methods. They propose a Gradient Shapley and a compound of hardness and Gradient Shapley as the data valuation method. Empirical investigation shows the comparison of these two proposed methods with several existing methods.

**Strengths:**

- The data valuation problem is highly relevant and interesting to the community. In particular, the issue being studied is the computational cost of Shapley value-based methods. It is also a highly relevant issue.
- The writing is relatively clear.
- There are experimental results against several existing methods.

**Weaknesses:**

- The authors propose a new utility function that takes into account the "hardness". From lines 234-235, it seems the optimization objective of the ML model is changed. However, this new optimization objective does not seem to have been theoretically justified.
- The proposed approach utilizes an approximation of the change in the loss function, but does not seem to show how good this approximation is.
- Furthermore, the proposed approach requires computing a per-sample gradient and storing these gradients. It does not seem to significantly reduce the computational cost of training, and additionally incurs memory overhead.
- More extensive expreimental results, larger datasets and larger models can add to the strength of the claims.

**Questions:**

How are (Wu et al., 2022) and (Ki et al., 2023) related to this work? Should they be compared?

`In this paper, we focus on the efficiency problem of data valuation on large-scale datasets.`

How large is considered large-scale datasets? Data-OOB seems to be able to scale to millions of data points. Is there comparison on datasets of similar scales?

What is $\alpha$ in Theorem 1?

What is the memory overhead from storing the per-sample gradients?

Typically, mini-batch SGD is used for training (namely the gradient is computed w.r.t. a mini-batch instead of each data point), which does not compute per-sample gradients. Does that mean you method needs to incur additional computational overhead?

What is the error (theoretical or empirical) of the last-layer gradient as an approximation?

_References_

DAVINZ: Data Valuation using Deep Neural Networks at Initialization. Zhaoxuan Wu, Yao Shu, Bryan Kian Hsiang Low. In ICML 2022.

DATA VALUATION WITHOUT TRAINING OF A MODEL. Nohyun Ki, Hoyong Choi, Hye Won Chung. In ICLR 2023.

---

> ### Author Response · Authors · 2024-11-23
>
> Thank you for your thoughtful review and questions. You seem to have a deep understanding of the data valuation field, and I appreciate your insightful inquiries.
>
> 1. **Large-Scale Datasets**: CIFAR-10 and CIFAR-100 are typically considered standard benchmark datasets for modern neural networks, especially in computer vision. However, even these datasets contain 60,000 samples. In my view, a dataset would be considered large-scale if it contains at least 50,000 samples. In Data-OOB, the largest dataset used contains only about 10,000 samples, and it appears that Data-OOB employs decision trees (with 800 trees in its experiments). I'm uncertain whether this method would perform well on larger datasets like CIFAR-100.
>
> 2. **Alpha in Theorem 1**: The parameter $\alpha$ in Theorem 1 can vary depending on the definition of the utility function. In this paper, it represents the mean gradient over the entire training dataset, but it could also represent the mean gradient over the validation dataset [1] or even the gradient of a specific sample [2].
>
> 3. **Computational Cost of Training**: Regarding the computational overhead, as shown in Table 2, the total training time is 2.2 hours. The times for different fractions (0.05, 0.1, 0.3, 0.5, 0.7) are 0.3, 0.6, 1.4, 1.7, and 2.2 hours, respectively. Using a 0.05 fraction significantly reduces the computational cost of training by about 10 times.
>
> 4. **Memory Usage**: Since we use the last-layer gradients, the memory usage for CHG Shapley is comparable to methods like Glister and GradMatch. For CIFAR-10, this results in a 5120 (512*10) dimensional vector, and for CIFAR-100 (which only uses biases), it results in a 100-dimensional vector.
>
> 5. **Mini-batch SGD**: If we treat each mini-batch as a player, my method behaves similarly to how per-sample are treated, just treat the gradient of a mini-batch as a gradient of a sample.
>
> 6. **Error in Last-Layer Gradient**: I am not entirely familiar with the theoretical or empirical error of using the last-layer gradient as an approximation. However, this approach has been widely adopted in various works like Glister and GradMatch, so it has shown to be effective despite the potential for some approximation error.
>
> 7. For some of your questions, I can provide answers, but others may remain unresolved in the short term, as these challenges are still under exploration in the academic community.
>
> Thank you once again for your insightful and thoughtful feedback. I truly appreciate the time and effort you put into reviewing this work, and I hope these clarifications help address your concerns.
>
> [1] Data Shapley in One Training Run
>
> [2] Estimating Training Data Influence by Tracing Gradient Descent

---

> > ### Comment · Reviewer_jhq7 · 2024-11-27
> > **Thank you for the response**
> >
> > I thank the authors for their response. Some details are provided for my questions. However, after reading through the response and the paper, I am not completely convinced of the contribution of this paper. It appears that this paper does not address an entirely new problem (see my listed references that also target this problem), the proposed method of using gradients and specifically last-layer gradient as an efficient approximation is not entirely new (the authors mention that such approximation is observed to be effective in existing works) and there does not seem to be new theoretical insights or results for this proposed method. While I appreciate the efforts that go into this paper, I would like to maintain my recommendation,

---

> ### Author Response · Authors · 2024-12-01
>
> Dear Reviewer jhq7,
>
> Thank you for your thoughtful comments and for taking the time to review our paper. I would like to address your concerns in more detail.
> 1. **Use of Last-Layer Gradient Approximation**:  We do not claim the last-layer gradient approximation as a novel contribution. As stated in the paper, we are "incorporating common tricks from data selection methods," including the last-layer gradient. Actually, this method has been used even in data attribution method like TracIN [2] and in its implementation of the influence function.
> 2.  **Literature Comparison**: I reviewed the references you provided and found their methods more restrictive than ours. The first relies on the Neural Tangent Kernel with an infinitely wide network, while the second focuses on two-layer overparameterized networks. We have cited these works in the **revised** manuscript. Additionally, our experiments use the open-source `opendataval` library, which does not support these methods, preventing direct comparison.
> 3. **Contribution to the Data Valuation Community**:
>    - **Motivation**: The goal of our work is not to solve a completely new problem but to address a practical and important challenge. We are inspired by the elegant work of Data Shapley, which combines game theory and machine learning to fairly measure the contribution of each sample. **We believe this approach deserves wider attention**. I aim to extend this work to the broader field of trustworthy machine learning, and I have addressed its major practical challenge: efficiency. In both data valuation and data selection tasks, we compare our method with the latest methods and achieve comparable, and in some cases, significantly better performance (see Table 4).
>    - **New Theoretical Results**: We believe our work offers a significant theoretical contribution. Equation 4 provides a novel theoretical result for the data valuation community and strictly follows the definition of the Shapley value. For exampe, to address your earlier concern about the role of $\alpha$, we clarify that if $\alpha$ represents the mean gradient over the entire training or validation dataset, CHG Shapley can be viewed as a data valuation method. If $\alpha$ refers to a specific sample, CHG Shapley becomes a data attribution method, similar to influence functions and TracIN.
>    - **Hardness in Data Valuation**: CHG Shapley also introduces the concept of "Hardness" to data valuation community. This is particularly useful in non-noisy scenarios (as demonstrated in Tables 2 and 3 with settings of 0.05 and 0.1). We show that GradE Shapley (a varient of CHG Shapley without Hardness term) is more effective in noisy data scenarios (as seen in Table 4), and incorporating Hardness into CHG Shapley reduces the performance of the trained model. While the role of Hardness has been explored in data selection, its application in data valuation has remained unclear. Our paper sheds light on this aspect and provides valuable empirical results.
> 3. **Contribution to the Data Selection Community**:
>    - **Data Valuation for Data Selection**: This paper is the **first** to apply data valuation-based data selection method on standard datasets like CIFAR-10 and CIFAR-100 (which are large-scale datasets for data valuation task). Previous data valuation methods were hindered by computational costs and could not perform these experiments. Our data valuation-based data selection method exhibits remarkable noise robustness—specifically, even when only 10%-30% of the data is selected, GradE Shapley can **outperform using the full dataset!** (see Table 4). And it significantly outperforms other existing data selection methods.
>    - **New Theoretical Insights**: In Theorem 2, we provide a theoretical explanation of why data valuation methods can be applied to data selection, which has been taken for granted in prior work. Theorem 2 offers a new probabilistic interpretation of the Shapley value: it represents the probability of a sample bringing a benefit to the existing selected subset.
> 4. **New Directions for the Data Valuation Community**:
>    Theorem 2 also points to a potential future direction for the data valuation community. Data selection and data valuation are, to some extent, **inverse processes**. And a central question arises: **When a sample is added to the training subset, how does the value of the remaining samples change?** Theorem 2 reveals that the Shapley value may change in this case, as the set of selected samples no longer follows a uniform random permutation. This could explain why CHG Shapley performs well at small fractions of data but struggles with larger ones. As we discuss in the "Discussion and Future Work" section, the solution lies in developing efficient, iterative methods to evaluate data value as new data points are selected.
>
> I hope this clarifies your concerns and better highlights the contributions of our paper. Thank you again for your valuable time.

---

### Official Review · Reviewer_hu3m · 2024-11-03

**Soundness:** 3
**Presentation:** 3
**Contribution:** 2
**Rating:** 3
**Confidence:** 3

**Summary:**

The paper studies data evaluation with regard to its contribution to model performance. Prior work on Shapley value and Data Shapley suffers from high resource and time requirements to evaluate each data point. To circumvent this issue, the authors propose instead to estimate the utility function using the CHG (Compound of Hardness and Gradient) score to obtain a close-formed Shapley value for each data point. This approach reduces the computation complexity of Data Shapley to that of a single model training run. Furthermore, the authors extend this framework to real-time data selection, which was previously impractical using prior methods of calculating data value.

**Strengths:**

- The proposed method of combining hardness and gradient utility function is novel.
- The authors provided experimental results to support their theoretical claims.

**Weaknesses:**

- The writing is confusing and lacks intuition. In Theorem 1, the authors provided a long equation 4 without explaining the importance of each term in this expression.
- In Table 2, it is not immediately clear why CHG Shapley outperforms other methods w.r.t accuracy and time. For example, CHG Shapley does not always provide the highest accuracy across different fractions of CIFAR 10 and CIFAR 100. Moreover, in CIFAR10, AdaptiveRandom sometimes even outperforms CHG Shapley while running for a shorter time.
- Similarly, in Table 3, the authors only draw the conclusion that CHG Shapley is outperforming Gradient Shapley. However, CHG Shapley is not better than existing methods in most comparisons.

**Questions:**

- Can the authors clarify how CHG Shapley is doing better than prior work when the empirical findings seem to suggest the contrary?
- Can the authors provide more intuition in Equation 4 (Theorem 1)? What does each term here mean and where do they come from?

---

> ### Author Response · Authors · 2024-11-23
>
> Thank you for your detailed review and feedback. Below, I would like to address the key concerns you raised:
>
> 1. **Explanation of Equation 4**:
>    The complexity of Equation 4 stems from its explicit consideration of the dataset size's influence on the gradient. For example, with $S$ samples, the gradient is $1/S \sum_{i=1}^S g_i$. When a single point $j$ is removed, the gradient should be expressed as $1/(S-1) \sum_{i=1, i\ne j}^S g_i$. This adjustment reflects the change in the direction of gradient desecnt when excluding a data point, which is also **why Equation (4) appears so complex**. Our paper rigorously incorporates this aspect, capturing the dependency of Shapley value of single data on dataset size, which is the key contribution of this paper and the distinction with [1].
>
> 2. **Discussion of Table 2 (Time consumption)**:
>    The time results in Table 2 highlight a significant milestone in data valuation research: this is the first time Shapley value-based methods have been made practical for data selection tasks on large-scale datasets like CIFAR-10 and CIFAR-100. Prior Shapley-based methods (listed in Table 1) typically require 100 times more computation time (see Table 6) and are impractical for datasets of this scale. CHG Shapley bridges this gap by achieving competitive results while maintaining computational feasibility, which is a key contribution of this work.
>
> 3. **Empirical Findings**:
>    We appreciate your careful examination of the experimental results. To clarify, **both** CHG Shapley and Gradient Shapley (which does not include the hardness term) are methods proposed in this paper and are derived using Equation 4. The detail distinction between CHG Shapley and Gradient Shapley is discussed in detail in Lines 242-253 and 493-498 of the paper. Specifically, CHG Shapley integrates the hardness term to improve data selection, which is very useful when selecting high-value data in non-noisy scenarios (see Table 2 and 3, settings of 0.05 and 0.1); Gradient Shapley is particularly well-suited for noisy data scenarios (Table 4) and incorporating Hardness in CHG Shapley actually decreases the performance of the trained model. While Hardness has been explored in the context of data selection [2], its role in data valuation remains unclear. This paper provides an empirical result that sheds light on this aspect, offering valuable insights into its application.
>
> 4. **Accuracy Issue**: As discussed in the "DISCUSSION AND FUTURE WORK" section, CHG Shapley performs well with smaller fractions but struggles with larger ones. This challenge may stem from the inherent difficulties of data selection based on Shapley values [3], rather than the limitations of CHG Shapley itself. Currently, the academic community lacks a clear understanding of how to address this issue effectively. Moreover, considering the computational demands of the LLM training process, the focus of both industry and academia is often on scenarios where the fraction of data selected is very small, such as 0.05 [4]. In these cases, CHG Shapley and Gradient Shapley generally outperform other methods across different datasets and settings.
>
> 5. **On AdaptiveRandom**: While AdaptiveRandom sometimes achieves better results at larger fractions (e.g., 0.3, 0.5, 0.7), it's important to note that selecting such large fractions (e.g., every 20 epochs in a 300-epoch training schedule) nearly covers the entire dataset. When the dataset is noise-free, this naturally leads to better performance. However, in noisy datasets, random selection can harm model performance, making AdaptiveRandom a poor choice in such scenarios.
>
>
> Thank you again for your thoughtful feedback, and I hope these clarifications address your concerns.
>
> [1] Data Shapley in One Training Run
>
> [2]  Infobatch: Lossless training speed up by unbiased dynamic data pruning
>
> [3] Rethinking Data Shapley for Data Selection Tasks: Misleads and Merits
>
> [4] LESS: Selecting Influential Data for Targeted Instruction Tuning

---

> ### Author Response · Authors · 2024-12-01
>
> Dear Reviewer hu3m,
>
> Thank you for your valuable feedback. To address the confusion regarding the method name, I have revised it, as suggested by another reviewer iunn, from "Gradient Shapley" to "GradE Shapley" (Gradient Euclidean Shapley). I hope this change clarifies the theoretical and practical consistency of the paper.
>
> Both CHG Shapley and GradE Shapley (which omits the hardness term) are methods **proposed** in this paper and are derived using Equation 4. As shown in Table 2 (standard setting) and Table 3 (class imbalance setting), CHG Shapley consistently outperforms GradE Shapley across **all** fraction ratios, emphasizing the importance of the hardness term in data evaluation. However, under the noisy label setting, GradE Shapley outperforms CHG Shapley and other previous methods across **all** fraction ratios. This is because the unreliability of the hardness term in the presence of label noise causes CHG Shapley to underperform relative to GradE Shapley.
>
> If you have any further questions, please do not hesitate to reach out.

---

> > ### Author Response · Authors · 2024-12-02
> > **Follow-up on Our Rebuttal**
> >
> > Dear Reviewer hu3m,
> >
> > We want to remind you that our rebuttals have been posted for a while, it's now approaching the end of the disscussion period. we are eager to hear your thoughts and any further feedback you might have.

---

### Official Review · Reviewer_iunn · 2024-11-06

**Soundness:** 4
**Presentation:** 4
**Contribution:** 1
**Rating:** 6
**Confidence:** 4

**Summary:**

The paper proposed a gradient-based method to reduce the computational cost of Data Shapley: the CHG (compound of Hardness and Gradient) utility function, which approximates the utility of each data subset on model performance in every training epoch. By deriving the closed-form Shapley value for each data point using the CHG utility function, they reduce the computational complexity to that of a single model retraining. They test CHG Shapley for real-time data selection in three settings: standard datasets, label noise datasets, and class imbalance datasets.

**Strengths:**

The paper is very well written and organized. The main idea and the method are presented very clearly.

**Weaknesses:**

I do not see substantial difference between the proposed method and the existing gradient based methods, especially the following one. Even in the experiments, the difference is not always noticeable.

- Wang et al., 2024, Data Shapley in One Training Run.

Minor comments: The previous gradient-based methods are not listed in Table 1.

**Questions:**

Is there fundamental difference between your method and the methods in the three papers below? What is the advantage of your method?
- Wang et al., 2024. Data Shapley in One Training Run.
- Xia et al., 2024. LESS: Selecting Influential Data for Targeted Instruction Tuning
- Pruthi et al. 2020. Estimating Training Data Influence by Tracing Gradient Descent

---

> ### Author Response · Authors · 2024-11-23
>
> Thank you for your detailed review and thoughtful comments. I’d like to address your concerns and clarify the distinctions:
>
> 1. **Difference with *Data Shapley in One Training Run* [1]**: The primary distinction lies in the approximation of the utility function. This paper uses Euclidean distance (and this is the first time), while [1] relies on a Taylor approximation. Although both approaches aim to derive an analytical Shapley value expression, the resulting formulations differ significantly. A notable limitation of [1] is its disregard for the influence of dataset size. For instance, with $S$ samples, the gradient should be expressed as $1/S \sum_{i=1}^S g_i$, but in [1], it is treated as $\sum_{i=1}^S g_i$.  In contrast, our paper strictly considers this aspect, which is also ***why Equation (4) appears so complex***. While the analytical approach in [1] is undoubtedly intriguing, it seems more suitable for inference scenarios rather than training. Additionally, [1] does not compare its method with other data valuation approaches (at least experimentally), as this paper does.
>
> 2. **Clarification on Table 1**: The connection between CGSV and Shapley value is primarily nominal. Methods like TracIn and LESS are unrelated to data valuation, as they do not calculate the value of data within the context of the entire dataset. Therefore, they are typically not considered data valuation methods (TracIn aligns more closely with data attribution, while LESS is geared towards data selection) and should not be included in Table 1.
>
> 3. **Advantages of CHG Shapley**: The method excels at identifying high-value data in data selection tasks (Table 2 and Table 3, settings of 0.05 and 0.1) and noisy data scenarios (Table 4). Experimental results demonstrate that CHG Shapley outperforms the other two gradient-based methods (Gradient-dot and Gradient-cosine, corresponding to the mentioned 4 paper by the reviewer) in these contexts. Furthermore, CHG Shapley is the second-fastest Shapley value approximation (after LAVA) among data valuation methods (Table 6).
>
>
> [1] Data Shapley in One Training Run

---

> > ### Comment · Reviewer_iunn · 2024-11-26
> >
> > Thank you for the explanations.
> > 1. Thank you for the clarification.
> > - Due to the sentence "It suggests that the Euclidean distance of two gradients is not just a similarity metric like inner product or cosine", I thought your square distance difference was somewhat equivalent to the inner product used in TracIn and LESS, because of the law of cosines b^2+c^2-a^2 = 2bc*cosine where cosine would just be the inner product used in TracIn and LESS. Now I understand that they are not exactly equivalent. But they still look similar. Could you further clarify the difference and why you want an upper bound of the training loss instead of just an approximation?
> > - Why is the sample size S so important? I believe all of the existing (gradient-based) methods are heuristic approximation of the Shapley value, including this paper. Does the sample size really make a difference in practice?
> > 2. I personally consider TracIn and LESS as data valuation methods. The idea of using gradient approximation was present in the earliest paper Ghorbani and Zou, "Data Shapley: Equitable Valuation of Data for Machine Learning" in **Algorithm 2 Gradient Shapley**. There is a more recent paper Evans et al. "Data Valuation with Gradient Similarity" that in some sense highlights the idea more explicitly.
> >
> > Another minor comment: I suggest changing the name of Gradient Shapley because it has been used in previous work.

---

> > > ### Author Response · Authors · 2024-11-28
> > >
> > > Thank you for your recognition of our work and your thoughtful considerations. Let me answer the question:
> > >
> > > 1. **Difference between Cosine, Dot and Euclidean Distance Similarity in Gradient-Based Methods**:  As you pointed out, the difference cosine, dot product, and Euclidean distance similarity can be subtle, and in practice, the choice of similarity metric often depends on experimental results (In our experiments, we found that CHG Shapley outperforms both Gradient-dot and Gradient-cosine in identifying high-value data and distinguishing noisy data scenarios.).
> > >
> > > 2. **Shapley Value under Different Similarity Metrics**:  Although the differences between the similarity measures are small, incorporating these into the **Shapley value computation** results in very different forms for each similarity metric:
> > >    - The dot product-based Shapley value still similar to the dot product form, though it requires subtracting a constant (see Appendix B, derivation of $ \phi_j(U_2) $).
> > >    - The Euclidean distance corresponds to the formulation in Equation (4).
> > >    - However, for cosine similarity, it is really hard to compute the Shapley value due to norm-related issues and I'm almost sure that it's impossible to derive a clean solution for this case. This is an open problem.
> > >
> > > 3. **Upper Bound of Training Loss vs. Approximation**:  The theoretical foundation of the paper aims to estimate the change in training loss when applying gradient descent to the current network parameters across different subsets of the training data. The **upper bound** of $ f(\theta_x) - f(\theta) $ serves as a **conservative approximation** of the training loss reduction (**conservative** means the upper bound holds for any $\theta_x$). The reason I didn't use an approximation is that I initially considered commonly used optimization bounds but hadn't thought of applying a Taylor expansion in this context :)
> > >
> > > 4. **Gradient Similarity as a Data Valuation Method**:  I totally agree that gradient similarity can serve as a heuristic for data valuation (and the mentioned papers indeed provide some evidence), but as pointed out on 2, I am cautious about framing this method as a strict data valuation technique because they do not directly compute Shapley values, as in the "Data Shapley: Equitable Valuation of Data for Machine Learning" paper. Therefore, to maintain theoretical rigor, I prefer to avoid calling it a "data valuation" method in the strictest sense though they have been used to measure the data value in practice.
> > >
> > > 5. **Why is the sample size S so important?**
> > >     - **From Machine Learning View**: The division $S$ actually represents the operation of taking the expectation. In traditional machine learning theory,  the average loss over  $S$  samples (i.e., empirical risk) approximates the generalization risk. This approximation is foundational for theoretical bounds in machine learning, such as Hoeffding’s inequality, which quantifies the relationship between empirical risk and generalization error. Practically, almost all machine learning algorithms compute the average loss over all available samples.
> > >     - **From Cooperative Game Theory view**:  The devision of sample size  $S$ affects the accuracy of Shapley value computations. Consider 5 samples (4 good, 1 noisy), with gradients $ g_i $ and the model update direction $ g_{\text{mean}} $. The gradient inner products are [0.6, 0.6, 0.6, 0.6, 0.1], where the noisy sample’s gradient is less aligned with the correct direction. Using a first-order Taylor expansion for the utility function $ U(S) = \sum_i \langle g_i, g_{\text{mean}} \rangle $, the Shapley values are [0.6, 0.6, 0.6, 0.6, 0.1]. In contrast,  using the rigorer form utility function as in this paper ($ U(S) = \frac{1}{S} \sum_i \langle g_i, g_{\text{mean}} \rangle $) results in Shapley values of [0.152, 0.152, 0.152, 0.152, -0.108]. Typically, negative Shapley values indicate low-quality samples (In the experiment shown in Figure 3 of [1], negative Shapley-value samples were also filtered out to get the "clean" datasets) , but the first method assigns 0.1 to the noisy sample, while the second assigns -0.1. This demonstrates that neglecting the sample size $S$ leads to inaccurate Shapley value estimation.
> > >
> > > 6. **Naming "GradE Shapley"**:  Thank you for pointing out the potential confusion with the term "Gradient Shapley," as it has been used in prior work. While Algorithm 2 in Ghorbani and Zou’s paper and our approach are conceptually quite different (the former affects model training process because of the Monte Carlo sampling of data subsets, while our method does not (see algorithm 1)), we recognize the need for clearer terminology. Given this overlap, I have revised the name to "GradE Shapley" (Gradient Euclidean Shapley) to better reflect the specific approach and avoid any confusion with the earlier work.
> > >
> > > Thank you once again for your valuable insights. Your suggestions help improve the clarity and rigor of our work.

---

### Meta-Review · Area_Chair_VE3J · 2024-12-20

**Metareview:**

The paper proposed a gradient-based method to reduce the computation of the Shapley value of training data. The proposed CHG (compound of Hardness and Gradient) utility function approximates the utility of each data subset on model performance.  Using the CHG utility function, the authors show that the task of computing data shapley reduces to a single model retraining.

Reviewers are most concerned about the novelty of the paper and unclear performance improvements compared to existing approaches. For the former, the authors might want to add discussions on how their paper related to several references raised by the reviewers. For the latter, the authors are encouraged to revisit their experiment results.

**Additional Comments On Reviewer Discussion:**

The primary concern on the paper's contribution remained after the rebuttal.

---

### Decision · Program_Chairs · 2025-01-22

Reject